# Effect of Pressure on the Removal of NH₃ from Hydrolyzed and Pre-Fermented Slaughterhouse Waste for Better Biomethanization

**Aleš Zver [1]**[ID]**, Rajko Bernik [2] and Rok Mihelič [2],***

[1]  KG Lendava d.d., Glavna ulica 115, 9220 Lendava, Slovenia; ales.zver@gmail.com
[2]  Biotechnical Faculty, University of Ljubljana, Jamnikarjeva 101, SI-1000 Ljubljana, Slovenia; rajko.bernik@bf.uni-lj.si
*  Correspondence: rok.mihelic@bf.uni-lj.si; Tel.: +386-40-795-561

**Abstract:** Slaughterhouse waste (SW) is potentially a good source of biomethane; however, its excessive ammonia content quickly causes inhibition of microbial processes. Our aim was therefore to remove ammonia from SW before putting it into a biogas reactor. Experimental 120 L pressure container was constructed to observe NH₃ removal from diluted slaughterhouse waste at constant air flow of 144 NL/min, temperature 130 °C, and at different pressures: 300 kPa, 600 kPa, and 900 kPa. SW was first allowed to hydrolyze for 14 days at 38 °C. The SW was diluted with water (DSW) to 8.4% dry matter (DM) and forcibly aerated for 334 min. From the DSW, 0.7%, 3.8%, and 9% of initial total N were removed at 300 kPa, 600 kPa, and 900 kPa, respectively. However, the C/N ratio changed only slightly, from the initial 4.38 to 3.17, which is not a promising result for biomethanization. Further research on the presented system with the addition of bases might be promising to remove more ammonia.

**Keywords:** slaughterhouse waste; hydrolyzes; pressure; pilot-scale; ammonia; biogas

---

## 1. Introduction

Organic waste can be an important source of new compounds, nutritional substances, and products such as biofuel, e.g., biogas. Slaughterhouse waste (SW), used as a basis for our research, is rich in protein [1]. Protein decomposition in a biogas reactor can be observed from the increase of volatile organic acids (VOA) and release of ammonium [2]. Ammonium's nitrogen inhibits the digestion process in high concentrations, especially at high pH of the digestion broth, when volatile ammonia (NH₃) is formed [3]. The total ammonium nitrogen content in a biogas fermenter is in the range from 1.7 to 14 g/L and becomes inhibitory at concentrations from 759 to 4000 mg N/L [4–6]. Especially during the digestion of pig and poultry waste, the high quantity of volatile ammonia frequently inhibits biogas production [7,8]. An increase in gaseous ammonia is caused by increased process temperature and the alkalinity of the environment [9]. Also, differences among substrates affect this inhibitory nature [9]. NH₄⁺ enters microorganism cells, causes proton imbalance, and runs counter to the metabolic enzymes in microorganisms [3,10]. During inhibition, the fermenting substrate starts accumulating VOA, which additionally decelerates other phases of digestion [3,11]. In the worst-case scenario, such inhibition can last for months and can seriously affect the economic outcome of biogas plants. This is why research is focused on balancing the ammonia content [11].

Various methods of removing ammonia from the wastes/mixtures before putting them into the biogas reactor have been tested. Ammonia can be removed via the method of air stripping with increased pH and temperature [12], with ion exchangers [13], by struvite precipitation [14], and with

biological processes [15]. Increasing the C/N ratio of input materials is also a good method of preventing inhibition by ammonia [1]. Dilution of ammonia-rich mixtures with water proved to be an effective and economically acceptable method in terms of stability and methane yield [16]; however, it can seriously lower the energy value of the substrate and, hence, the production of biogas [17]. The inhibitory effect of ammonia can be diminished also by adding ion-exchange capacity materials, such as bentonites, glauconites, and phosphorites, that can bind nitrogen in its ammonium ($NH_4^+$) form [18,19]. The use of specialized microbial associations and adaptation of the existing microbial associations can ease the ammonia burden [20]. Ammonia removal can also be done with the use of membranes [21] by affecting the balance of $NH_3$ and $NH_4^+$ by changing the pH and temperature. $NH_3$ was dissociated through the pores of the membranes sized between 0.1 and 5 μm [22,23]. Recently, an informative review on the development in ammonia stripping process was published by Kinidi et al. [24].

For the biogas fermentation of high-ammonia substrates, thermophilic fermentation at 55 °C proved to be a better option as it did not result in such high inhibition as in the mesophilic process. Temperature is of crucial importance as a higher temperature facilitates increased removal of $NH_3$ from liquid to biogas [23–25].

Although various methods have been researched and tested, there is to our knowledge, a lack of research on ammonia removal with increased pressure from hydrolyzed and pre-fermented slaughterhouse waste. The concentration of a solute gas in a solution is proportional to the partial pressure of that gas above the solution and the solubility of gases generally decreases with increasing temperature (Henry's law). Hence, higher pressure would lower the volatility of ammonia. However, free ammonia formation is not just partial pressure dependent. Ammonia release is also very much pH dependent: The volatility of ammonia in an aqueous solution can be enhanced by increasing the pH [26].

$$[NH_3]_{aqueous} + [H_2O] \leftrightarrow [NH_4^+] + [OH^-]$$

High pressure can also help hydrolysis of proteinaceous material and release of soluble ammonium N, which would further enhance the possibility of N removal from slaughterhouse waste. Our contribution is therefore an experiment of a novel approach to ammonia removal for nitrogen-rich waste. For this purpose, we made a pilot-scale experimental pressure chamber. Our hypothesis is that the system presented in this study will effectively remove ammonia from pre-hydrolyzed SW and thus increase the C/N ratio of the SW, which would both render the mixture more suitable for anaerobic biogas fermentation and increase methane yield.

## 2. Materials and Methods

### 2.1. Hydrolysis and Fermentation

SW was dried and ground to a powder with particle size up to 1.5 mm. A Bremey 551.0 grinder (Type 31; Meyer Mühlenbau AG, Solothurn Switzerland) was used for grinding to produce a homogeneous input for further treatment.

The SW was diluted with water to obtain the percentage of dry matter (DM) in the mixture that would allow easy transport via pipelines by means of pumps. Accordingly, diluted SW (DSW) made up of 60 g of SW moist paste with 53.2% DM (Figure 1) and 300 g of $H_2O$ was set for five repetitions. In terms of volume, this represents 16.7% SW and 83.3% $H_2O$. The DSW contained 8.4% DM and 6.8% ODM. An automatic methane potential test system (AMPTS), Bioprocess Control AB, Sweden, was used in the laboratory (bench-scale) trial. Each mixture was then put into a fermenter with a 0.5 L volume and placed in a water bath at 38 °C. The content was mixed at a frequency of 15 revolutions per minute; an automatic stirrer was switched on every 5 min. The anaerobic fermentation lasted for 32 days. During this period, the total quantity of the generated gas was measured by the AMPTS system. The measurements of $CO_2$ and other gasses were not performed since we removed the $CO_2$ traps from AMPTS system.

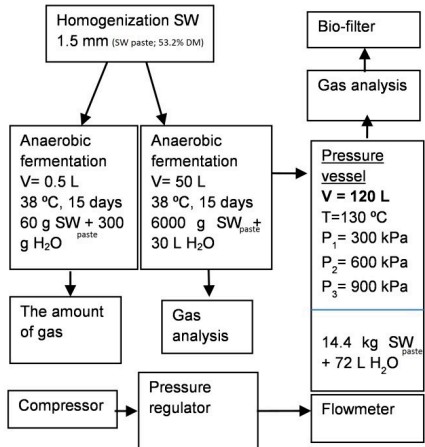

**Figure 1.** Diagram of the nitrogen removal trial. Homogenized slaughterhouse waste (SW) was first mixed with some water to obtain SW paste which was further diluted with water to get mixture containing ca. 8% DM. Diluted SW was anaerobically hydrolyzed and fermented in pre-trials (0.5 L and 50 L containers), and then in the pilot size 120 L pressure vessel, where ammonia was removed under three different pressures (300, 600, and 900 kPa), at constant temperature (T = 130 °C), and air flow (144 NL/min).

The entire experimental processes are shown in the diagram in Figure 1.

The need for a larger quantity of the anaerobically fermented mixture prompted us to mix the same input ratio increased by 100 times, i.e., 6 kg of SW paste and 30 kg of $H_2O$ put in a 50 L container. This was done simultaneously with the laboratory pre-trial (Figure 1). The mixture in the container was waterproofed and placed in a water bath at 38 ± 0.5 °C. A special stirrer was used to mix it thoroughly for 5 min three times per day. Once per day, a sample was taken by means of a special probe which prevented air from entering the container during sampling. The generated gases were then channeled to a bio filter via a pipe. Once per day, the gases were analyzed by means of a portable gas analyzer (Dräger: X–am 7000; Dräger Slovenija d.o.o., Ljubljana, Slovenia). The following parameters were set for the following methods: infrared sensor (IR) EX sensor for the measurement of methane (measurement range 0–100% by volume); IR $CO_2$ HC sensor for the measurement of carbon dioxide concentrations (measurement range 0–100% by volume); electro-chemical (EC) $O_2$ sensor for the measurement of the oxygen content (measurement range 0–25% by volume); EC $H_2S$ HC sensor for the measurement of hydrogen sulfide (measurement range 0–200 ppm); EC $NH_3$ sensor for the measurement of ammonia (measurement range 0–2000 ppm).

### 2.2. Overpressure Ammonia Removal

After hydrolysis, the DSW was placed in a specially adapted pressure vessel with a 120 L volume (Figures 1 and 2) with integrated nozzles that that provide the incoming air, which then mixes the content, and allow overpressure ammonia expulsion.

The vessel has three openings on the lid. A manometer is attached to one of them, another lets out the outgoing air, while the third one is intended for the measurement of temperature. On the sides, the vessel contains an opening for sampling and the attached nozzles. The vessel has two layers. The inner vessel was made to resist high pressure up to 2000 kPa, while the outer vessel is intended for pressure of 500 kPa. Both areas have a relief valve that protects them from the overload. The outer part is intended for a thermal-oil bath with three 1000 W electric heaters submerged in the oil. The thermostat in the layer regulates the temperature of the layer, which then heats the inner vessel. Overpressure was created by means of a NU AIR B2800/100 CM3 V230 compressor, (Nuair, Torino, Italy) and the air was channeled via pipes into the three nozzles in the reactor. The pipe was equipped with a non-return valve that prevented the air or mixture from returning to the compressor. The air stripping was regulated with the valve. It also enabled us to separately regulate the air flow to an

individual nozzle. The pressure of the incoming air was regulated with the pressure regulator located on the compressor. The quantity of the incoming air was measured with an air-flow measuring device (Side-Trak Mass Flow Meter, model: 830M-2-OV1-SV1-V4-HP; https://www.sierrainstruments.com/).

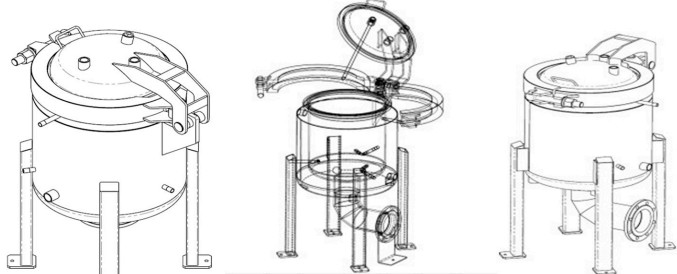

**Figure 2.** Pressure vessel (120 L) with overpressure ammonia stripping nozzles constructed for the purpose of the experiment. The inner vessel resists up to 2000 kPa. Outer layer contains thermal-oil bath with three 1000 W thermostatically regulated electric heaters, which heats the inner vessel. Diluted slaughterhouse waste was heated to 130 °C and air stripped with constant air flow of 144 NL/min under selected pressures, 300, 600, and 900 kPa.

The content—DSW—was air stripped at different pressures in the vessel (300 kPa, 600 kPa, and 900 kPa), while the air flow rate was constant at 143.8 NL/min and the temperature remained 130 °C during the entire experiment (Figure 1). The only variable was the pressure. The air stripping lasted for 334 min. The air passing through the vessel also removed ammonia, which was measured in the outgoing air by means of a gas analyzer (ECHO) with inbuilt EC $NH_3$ sensors [27] and is expressed as a volume ratio (mg/L).

*2.3. Other Measurements*

Measurements of total solids (TS), volatile solids (VS), and pH were done according to the American Public Health Association APHA standard methods for the examination of water and wastewater [28]. VOA and total inorganic carbon (TIC) were determined using the TIM 840 titrator from HACH LANGE (Hach Lange d.o.o., Domžale, Slovenia) following the instructions of the manufacturer. For the measurements of total Kjeldahl nitrogen (TKN) and ammonium nitrogen ($NH_4$-N and free $NH_3$), we used a modified Berthelot reaction; after organic nitrogen digestion, free ammonia is produced by high pH. Dialysis is then used to capture the released ammonia in a buffered and chlorinated stream. With salicylate addition, 5-aminosalicylate is formed. TKN and ammonium-N were measured using a SAN++ Continuous Flow Analyzer (Skalar Analytical B.V., The Netherlands (https://www.skalar.com/) [29]. The gas composition was determined using a Dräger X–am 7000. Measurements of an individual parameter were performed with three repetitions (*n* = 3). Cadmium, chromium, nickel, and lead were determined according to SIST EN 14083 [30] and mercury according to ISO 16772 [31], while calcium, magnesium, potassium, sodium, copper, iron, manganese, and zinc were determined according to ISO 6869 [32]. Crude fiber was determined according to ISO 5983-2 [33], crude protein according to ISO 6865 [34], and crude fat according to Commission Directive 98/64 [35], while the gross energy (calorific) value was determined according to CEN/TS 15400 [36]. The C and N were determined by means of a Vario MAX CN instrument working on the principle of combustion according to SIST ISO 13878 [37]. The C/N ratio was calculated from these measurements.

**3. Results**

*3.1. Changes During the Hydrolysis/Anaerobic Fermentation*

The SW analysis, which served as the trial basis, is shown in Table 1. The SW was dry (95% DM) and contained 83% ODM; its gross energy value was 22.56 kJ/g SW. The material was slightly acidic

($pH_{(H2O)}$ = 6.11). The main constituent was crude proteinaceous matter (63%). The content of fats was about 15%, whereas the amount of fibers was small (<3%). It was rich in minerals (crude ash content ca. 16%), of which Ca and P were the main elements. The level of potentially toxic elements like Ni, Pb, Cd, Zn, Cu, Hg, and Cr was low. The level of N was very high (10.1%), giving the SW a very low C/N ratio of 4.4; this is unfavorable for anaerobic fermentation to biomethane, as the optimal ratio is between 25 and 35 [38].

**Table 1.** Analysis of dry SW.

| Slaughterhouse Waste | Unit | Result | Slaughterhouse Waste | Unit | Result |
|---|---|---|---|---|---|
| Dry matter | g/kg | 953 | Na | g/kg | 4.81 |
| Moisture | g/kg | 47.2 | Mg | g/kg | 1.52 |
| Crude protein | g/kg | 632 | Ca | g/kg | 51.8 |
| Crude fiber | g/kg | 27.6 | Ni | mg/kg | <1.00 |
| Crude fat | g/kg | 154 | Pb | mg/kg | <5.00 |
| Crude ash | g/kg | 155 | Cd | mg/kg | <0.10 |
| Gross Energy value | kJ/g | 22.57 | Fe | mg/kg | 214 |
| pH | | 6.11 | Zn | mg/kg | 89.0 |
| C/N | | 4.40 | Mn | mg/kg | 10.0 |
| N | g/kg | 101 | Cu | mg/kg | 8.00 |
| P | g/kg | 26.7 | Hg | mg/kg | <0.40 |
| K | g/kg | 6.10 | Cr | mg/kg | <5.00 |

The SW was mixed with water to get diluted SW (DSW: 1 part SW paste to 5 parts $H_2O$; w/V; Figure 1), and allowed to hydrolyze and ferment in anaerobic conditions described in the Section 2.1. During the hydrolysis and fermentation, changes in the pH, DM, ODM, TIC, VOA, VOA/TIC, $NH_4$-N, total N, and energy value of the input were monitored (Table 2). In the SW, nitrogen was present predominantly in proteins. For this reason, SW was first hydrolyzed to disintegrate proteins into $NH_4$-N and fats into VOA. After 30 days, the process was almost complete; this was confirmed by the measurements of pH, VOA, and TIC (Table 2, Figure 3). After 15 days, the production of VOA reached nearly a plateau (Figure 3). That is why a 15-day period of anaerobic fermentation was set for putting the hydrolyzed DSW from the pilot scale hydrolysis, 50 L containers, into the pressure vessel (120 L) for ammonia stripping. The fresh DSW mixture contained 21,746 mg/L of TKN and 189 mg of $NH_4$-N (Table 2). After a 15-day hydrolysis and fermentation period (Figure 3), 8788 mg/L of $NH_4$-N and 23,500 mg/L of VOA were generated in the mixture, which might have caused complete inhibition of methanogenesis; there was no $CH_4$ in the outgoing gases. The ammonia inhibition is discussed later. In addition, the generated quantity of $NH_4$-N also inhibited hydrolysis, but not completely, as a slow increase of VOA was still present (Table 2).

Gas was emitted during the hydrolysis and fermentation. In the 15-day period, the mixture emitted 16.66 ± 0.49 NL gas per kilogram of dry sample containing 32% $CO_2$, 0% $O_2$, 0.5% $CH_4$, 1500 μL/L $H_2S$, and 1084 μL/L $NH_3$. Furthermore, the outgoing gas contained other volatile compounds that emitted a very strong smell. The remaining gas in the outgoing air had to be mainly composed of $N_2$ (Figure 3).

The VOA content increased to 23,501 ± 125 mg/L during the 15-day hydrolysis and fermentation period. After 15 days, the increase in the VOA content levelled out. A similar trend was observed for pH, which also did not significantly increase after 15 days. It was interesting that the pH went from slightly acidic and stabilized at neutral reaction in spite of the high VOA production. The organic acid production was continuously neutralized by ammonium and TIC production during the degradation of organics, which both caused alkaline reactions. An increase in the TIC content decreased the VOA/TIC ratio after 15 days of hydrolysis and fermentation according to the reduced trend of VOA production. In 15 days, due to the released gases, the mixture was energetically just slightly impoverished by 393 kJ per gram of SW DM or 1.77% based on the initial gross energy value of DSW (Table 2).

**Table 2.** Changes in the diluted SW (DSW) during the hydrolysis and fermentation and the follow-up ammonia stripping under different pressures.

| Measurement | Unit | Fresh DSW | DSW After 15-Day Hydrolysis | 300 kPa Ammonia Stripping | 600 kPa Ammonia Stripping | 900 kPa Ammonia Stripping |
|---|---|---|---|---|---|---|
| Gross energy | kJ/g DM | 22.26 ± 6.0 | 21.87 ± 1.1 | - | - | - |
| pH | | 7.26 ± 0.1 | 6.86 ± 0.01 | 6.08 ± 0.02 | 5.55 ± 0.02 | 5.85 ± 0.01 |
| TIC | mg/L | - | 7627 ± 182 | 4920 ± 174.2 | 8879 ± 187 | 4918 ± 147 |
| VOA | mg/L | - | 23,501 ± 126 | 28,840 ± 247 | 46,697 ± 207 | 30,781 ± 198 |
| DM % | % | 8.4 ± 0.12 | 8.81 ± 0.22 | 7.88 ± 0.30 | 7.55 ± 0.44 | 7.35 ± 0.17 |
| ODM % | % | 6.8 ± 0.20 | 6.62 ± 0.15 | 6.63 ± 0.4 | 6.40 ± 0.43 | 6.28 ± 0.37 |
| ODM loss | | | | | | |
| C/N | ratio | 4.38 ± 0.04 | 4.22 ± 0.06 | 3.27 ± 0.02 | 3.42 ± 0.05 | 3.17 ± 0.03 |
| N total | mg/L | 21,746 ± 3 | 15,546 ± 113 | 14,437 ± 134 | 13,687± 137 | 12,683 ± 90 |
| $NH_4$-N | mg/L | 189 ± 39 | 8789 ± 39 | 8038 ± 114 | 6694 ± 130 | 5887 ± 111 |
| $NH_4$-N/N total in DSW | % | 0.9 | 56.5 | 55.7 | 48.9 | 46.4 |
| Stripped $NH_3$-N | mg/L | | | 98.0 | 519.7 | 1138.1 |
| Relative N removal: | | | | | | |
| Stripped $NH_3$-N/N total | % | | | 0.7 | 3.8 | 9.0 |
| Stripped $NH_3$-N/$NH_4$-N | % | | | 1.2 | 7.8 | 14.2 |

± value = standard deviation.

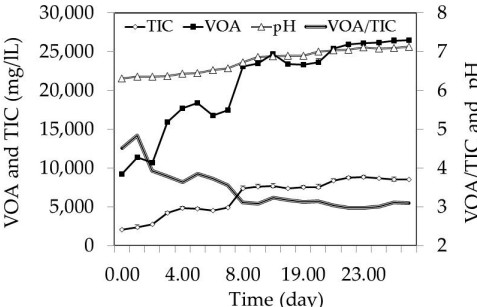

**Figure 3.** Changes in the DSW mixture during hydrolysis and fermentation. Values for total inorganic carbon (TIC), volatile organic acids (VOA), and pH were augmenting while the ratio VOA/TIC declined during 30 days of hydrolysis and fermentation.

*3.2. Overpressure Ammonia Stripping*

The main aim of our trial was to remove ammonia at 130 °C and at fixed pressures: 300 kPa, 600 kPa, and 900 kPa. By this we followed and exceeded the legal requirements that SW should be pre-treated at 130 °C and at 300 kPa pressure for at least 20 min [39].

We succeeded to remove only 98.0 mg/L, 519.7 mg/L, and 1138.1 mg/L $NH_3$-N or, expressed in relation to the initial TKN, 0.7%, 3.8%, and 9% at 300 kPa, 600 kPa, and 900 kPa, respectively (Table 2). Relatively inefficient ammonia removal was probably caused by neutral pH in the hydrolyzed and fermented mixture. Although the alkali forming $NH_4$-N was high after hydrolysis (8800 mg/L), a neutral to slightly acid DSW was probably caused by high concentration of VOA (23,500 mg/L; Table 2). The pressure treatment further increased VOA to max. 47,000 mg/L at 600 kPa. In the stripping process pH of DSW decreased from initial 6.9 to 6.1, 5.6, and 5.9 at 300 kPa, 600 kPa, and 900 kPa, respectively. This pH decrease was caused by VOA formation and by the removal of $NH_3$ with the exhausted air (Figures 4 and 5 and Table 2).

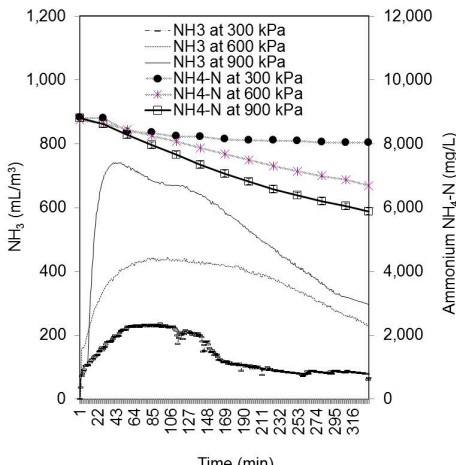

**Figure 4.** Ammonia stripping process: quantities of $NH_3$ (µL/L) in the exhaust air (lines) and quantity of $NH_4$-N (mg/L) in the DSW mixture (lines with marks). The ammonia gas ($NH_3$) concentration in the exhausted air raised fast in the first hour after pressure increase, and later slowly declined. The ammonium nitrogen ($NH_4$-N) content of the DSW linearly declined over 334 min of the process.

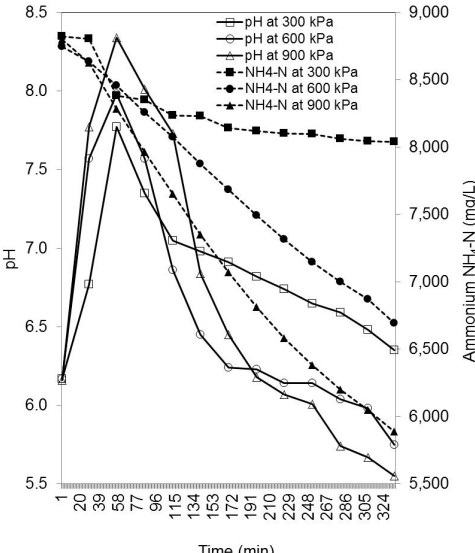

**Figure 5.** Relationship between pH (lines with open symbols) and $NH_4$-N (lines with solid symbols) in the DSW during ammonia stripping within a 334 min timeframe at different pressures. Overpressure caused firstly abrupt short-term alkalization of DSW with pH peaks at 60 min after the start of pressure and stripping, but later the DSW acidified again. The ammonium nitrogen ($NH_4$-N) content of the DSW linearly declined over 334 min of the process.

The high VOA content (and consequently lower pH) and high ammonia inhibited the methanogenic process, which is why the outgoing gases during hydrolysis did not include methane. At the beginning of the stripping process (the end of hydrolysis and pre-fermentation), the VOA content was 23,501 mg/L. It increased during the stripping process and reached its peak of 46,697 mg/L at 600 kPa pressure; it was lower at 300 kPa (28,840 mg/L) and at 900 kPa (30,781 mg/L). The VOA content is reflected by the pH level drop at each pressure (Table 2; Figure 5).

The latter proves that, in addition to ammonia, higher pressure also promotes the removal of carbon-containing compounds ($CO_2$ and other volatile organic compounds, such as VOA), which is reflected in the drop of the C/N ratio, from $4.38 \pm 0.04$ to $3.17 \pm 0.03$ (at 900 kPa pressure) (Table 2). Thus, the higher the pressure in stripping, the lower the C/N ratio at the end.

At the beginning of the stripping process, after 60 min of pressure stripping, pH rose up to 8.35 and then started decreasing (Figure 5). The initial increase in pH might be a result of the loss of the short-chain VOA and simultaneous ammonification process.

During the stripping process the concentration of VOA increased during ammonia pressure removal, while the TIC slowly decreased (Table 2). The increased concentration of VOA, and ammonia removal were reflected in a gradual decrease of pH (Figure 5); the pH decreased almost on a linear basis relative to the removed $NH_3$ (Figure 5). The lowest pH occurred at 600 kPa, which was linked to the highest increase of VOA content.

## 4. Discussion

Our idea was that by the proposed procedure nitrogen would be effectively stripped, rendering energy rich DSW, with substantially lower N content, which would allow higher loading rates of slaughterhouse waste in biogas reactor. Adjustment of $NH_4$-N and free ammonia content of input material (like DSW) into biogas plant is one of the important parameters in basic process control for successful biogas plant operation [40]. Volatile ammonia ($NH_3$) can be a strong inhibitor of anaerobiosis above threshold concentrations. The majority of research found the toxic level of total ammonium nitrogen for methanogenesis at around 1700 mg/L with non-acclimated inoculum, while with acclimation, no inhibition was apparent up to 5000 mg/L. If the pH is controlled bellow 8.1, the tolerable limit of $NH_4$-N is even as high as 6000 mg/L [41]. Under the same conditions, the toxic level of free ammonia ($NH_3$) appeared already at ca. 200 mg/L; if, however, the substrate was rich in soluble organic matter, and microbes were acclimatized, the ammonium toxicity appeared at concentrations from 400 to 800 mg/L [40,41]. In our experiment, the total ammonia ($NH_4$-N) level at the end of hydrolysis was at a very high level of 8800 mg/L. Immediately after starting to raise the pressure to 300, 600, or 900 kPa, the free ammonia levels raised abruptly from nearly zero to 750 mg/L, 400 mg/L, and 200 mg/L, respectively (Figure 4). The ammonium-N level dropped significantly during the pressure removal period (6 h), but even at the highest pressure, 900 kPa, where the ammonia removal was the most successful (Figure 5), it stayed at a high level (5800 mg $NH_4$-N/L) at the end of the procedure. These high levels could probably cause inhibition of methanogenesis, although free ammonia was not so abundant, as the pH was not very alkaline (the highest was up to 8.35 at 900 kPa). The alkalinity lasted just for a short period of ca. 2–3 h; after that time, the production of VOA lowered the pH and prevented further ammonia volatilization (Figure 5). The reduced pH of the DSW during ammonia stripping caused a loop: the lower the pH, the lesser the generation of $NH_3$. Regardless of the pressure used in the treatment, the ammonia removal was greatly reduced when the pH dropped below 6.0 (Figures 4 and 5). In general terms, the stripping of the hydrolyzed SW mixture did not achieve all of the pursued objectives. Nitrogen was partially removed, which was our goal. However, more C was lost during the process, which rules out our initial prediction that the process would optimize the C/N ratio and improve the ammonia-cleared input for digestion all the way to methane.

Other researchers stripped ammonia from anaerobically metabolized digestate at lower temperatures and at only slightly increased pressure but at changed pH levels with various alkaline compounds. In his trial, Guštin [42] alkalized the digestate with NaOH treatment to pH 12, which resulted in 88.3% removal of TKN and 96.8% of $NH_4$-N from the digestate at a continuous air flow of 2.5 L/s. In our case, with a very similar air flow rate (143 L/min = 2.38 L/s) but at pH below 8.0 for most of the stripping time, the nitrogen removal was only on the level of 0.7%, 3.8%, and 9% of the initial total N level at 300, 600, and 900 kPa pressure, respectively.

During ammonia pressure removal, the concentration of VOA increased to very high levels compared to reference data of mixed slaughterhouse waste digestion with similar initial total solids and volatile solids composition. Water evaporation, which might be the cause for augmenting of VOA concentration, was not directly measured, but since dry matter content was also decreasing during the process, we can deduce that water evaporation was not high. In our case, VOA levels increased up to 47,000 mg/L (at 600 kPa; Table 2) compared to 10,000 mg/L in the experiment by Ortner et al. [43], who

succeeded in lowering the level of VOA by addition of trace nutrients such as Ni, Co, or Mo. In our case, the Ni content was low (<1 mg/kg; we did not analyze SW for Co and Mo), so consideration of these trace elements' concentrations in SW should be made in the future development of our method. Such extremely high VOA levels consumed the buffer capacity of DSW, which was observed from the gradual decreasing of TIC. After 3 h of pressure stripping, the pH of the DSW dropped below 6.5, which is sub-optimal for methane production. The pH fell to as low as 5.5 after 6 h (Figure 5). The addition of buffer material (bicarbonates) could prevent acidification but can also prevent free ammonia ($NH_3$) accumulation by forming carbamic acid ($H_2NCOOH$) and carbamates ($2\ NH_3 + CO_2 \rightarrow NH_4[H_2NCO_2]$) [43].

## 5. Conclusions

In this experiment, we tested the possibility of removing excess ammonia from N-rich, hydrolyzed, and DSW by air blowing using different overpressures. It was proven that an increase in pressure from 300 kPa to 900 kPa positively influenced the removal of ammonia from the mixture; however, the removal was very low compared to that by, e.g., alkaline processes. In addition to $NH_3$, other volatile compounds were also removed; these are to be determined in detail by further research. The loss of volatile substances further narrowed the C/N ratio, which was already too low in the initial substrate (DSW) for optimal biogas digestion. For further development of the here-presented method of ammonia removal from N-rich biogas substrates, pressure could be combined with moderate substrate alkalization.

**Author Contributions:** Conceptualization, A.Z. and R.M.; Methodology, A.Z., R.M., and R.B.; Validation, A.Z., R.M., and R.B.; Formal Analysis, A.Z., R.M., and R.B.; Investigation, A.Z. and R.M.; Resources, A.Z.; Data Curation, A.Z., R.M. and R.B.; Writing—Original Draft Preparation, A.Z. and R.M.; Writing—Review and Editing, A.Z. and R.M.; Supervision, R.M. and R.B.; Funding Acquisition, A.Z. and R.M.

**Funding:** This research was funded by European Union, European Social Fund. The experiment was implemented in the framework of the Operational Program for Human Resources Development for the Period 2007–2013, Priority axis 1: Promoting entrepreneurship and adaptability, Main type of activity 1.1: Experts and researchers for competitive enterprises. The present article was written as part of doctoral dissertation which was financed by the Ministry of Higher Education, Science and Technology of the Republic of Slovenia and from the European Social Fund of the European Union.

**Acknowledgments:** We would like to express our gratitude and appreciation for the co-financing and research grant. The authors acknowledge the support of the companies Keter Invest, Keter Organica, and University of Ljubljana, Biotechnical Faculty for access to laboratories and research facilities.

**Conflicts of Interest:** The authors declare no conflict of interest. The funders had no role in the design of the study; in the collection, analyses, or interpretation of data; in the writing of the manuscript; or in the decision to publish the results.

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
