# Peer review of "Effect of Pressure on the Removal of NH3 from Hydrolyzed and Pre-Fermented Slaughterhouse Waste for Better Biomethanization"

_energies, doi:10.3390/en12101868_

Round 1
Reviewer 1 Report
General: 1.) Language needs a careful attention. Please take your time to improve it. Abstract: 1.) I would be more careful in what regards the aim of the study. As the title says, is the effect of pressure on the removal and not the remove the ammonia per se; Introduction: 1.) Seems to be OK. Pay attention to English; Materials and methods: 1.) Section 2.1.: try to shorten the title and to exclude symbols such as “=”; 2.) Line 80: The content was mixed with a frequency of 15 minutes. Please reformulate this. 3.) Line 87: 2.2. Hydrolysis/fermentation in a pilot scale trial. Hydrolysis and fermentation…? 4.) Line 120: 2.4. Experimental pressure vessel: Remove “:” Results: 1.) Line 180: try to find another way of reporting the note of the table: “± = standard deviation”; 2.) Line 181: it seems that is this may be a subtitle. Try to reformulate: “Gas was emitted during hydrolysis and fermentation.”; 3.) Figures need to be self-explanatory. Add descriptions for the features included in the figure caption; Discussion: 1.) Seems to be OK; Conclusions: 1.) Seem to be OK.Author Response
Dear Reviewer, we attach the word file with responses to your points.
Thank you and best regards, Rok Mihelič

Reviewer 2 Report
The authors investigated ammonia removal from slaughterhouse digestate with a device that works under high temperature and pressure. The topic is interesting and it can be an added value to the present literature. However, the authors failed the highlight the novelty of their work. In my opinion, the novelty of manuscript is about the ammonia stripping under high temperature and pressure. The authors decided to focus on the effect of ammonia on anaerobic digestion while they didn't perform any experiment with digestate after stripping. This creates an unnecessary discussion in the paper. In addition, the results of the pre-trial experiments do not contribute significantly to the take-home message of the manuscript. It is not a good idea to expect good AD performance under such conditions. It is documented in the literature as well. The authors should either clarify this part better or give this information as supplementary material. Therefore the authors should rewrite their manuscript, focusing on digestate stripping. This will increase the value of the paper. My specific comments are:
> The introduction part is extensively focusing on ammonia inhibition. while the effect of ammonia after ammonia stripping was not investigated. Therefore introduction should be more focused on ammonia stripping studies.
L-56-59 Incoherent sentence. Please resolve.
L 61 you need to mention what is known about stripping under high pressure.
> The working principle (hypotheses). should be integrated into the introduction.
L 68 what is phase 2? The authors define phase 1 but no info available for phase 2
L74-77 This part is not clear. I do not understand why the defined DS content was chosen.
L 82 Do the authors mean CH4?
L159 The authors should mention how caloric values were calculated.
L 165-166 This sentence belongs to the introduction.
L 181 - 184 . for gas values it is better to express them in ppm.
L204-205 I have some doubts about this statement. Air stripping should also remove some portion of volatile fatty acids. It might be another reason for increased concentrations.
L244- 256 this part is not relevant discussion for the paper since there is no experiment done after stripping. It must be discarded.
> The authors should introduce nitrogen mass balances into their manuscript. I strongly believe in such a system there can be water evaporation problems. This may explain increased VFA concentrations in the system. Nitrogen mass balance is very important for a successful evaluation of the stripping systems.
Author Response
Dear Reviewer, please find attached the Word document with answers to your comments. Thank you for your important input. Sincerely, Rok Mihelič

Reviewer 3 Report
This manuscript concerns a crucial problem in the methanisation process, that is the NH3 inhibition. Authors evaluated the potential of positive pressure to strip NH3 from the liquid after a hydrolysis step. Results do not show real positive effects, but it is important that such “negative” results could be published and put the knowledge to the scientific community. However, before publication, some questions or lacks must be answered. The manuscript should also be revised by an English-native person.
Line 37: NH3 (and not NH4+) enters micro-organisms cell diffusively
Line 83: in the AMPTS device, the CO2 trap removes this gas from the gas flow and therefore CO2 is not counted by the AMPTS; however, authors said that the measurements of CO2 were performed by using the CO2 traps. How did they do that?
Figure 2: in the left box, authors measured the amount, but also the nature of gas. That must be added.
Line 159: authors said that SW contains a lot of energy, more or less 22.5 kJ/g; this is not “a lot”, this is just normal considering that the calorific value of protein and lipids are more or less 16 and 40 kJ/g respectively.
Table 1: first of all, I don't think that all measurements of metals are really useful. Secondly, the calorific value must be given in kJ/g, which is more usual.
Table 2: some results presented in this Table, that are discussed later in the text, need more explanations.
First, calorific value (that should be expressed in kJ/L while it is DSW and not crude SW; and also they are not “Calories”) : how authors can explain the important drop after the pressure stripping. For instance, after hydrolysis the calorific value is about 22 kJ/mL, after 300 kPa only 16 kJ/mL. That means that more or less 21 % of the energy, i.e. molecule, were disappeared. However VOA increased, could authors explain?
Second, VOA : comparing after the hydrolysis step, and after the 300 and 600 kPa experiments, VOA goes from 23.5 to 28.8 and 46.7 g/L. How authors can explain that a pressure treatment can create VOA?
Thirdly, DM : the fresh DSW contains 16 % DM and after the hydrolysis only 8.8 % remains, what is the DM lost?
Line 184: did authors check a potential H2 production during the hydrolysis step?
Figure 3: the NH3 concentrations in the figure seem to be calculated, probably with f the TAN values and pH? All results are available and it is quite usual way to do. If true, authors should detail the formulas and the calculations.
Line 223-225: authors said that the higher pressure promotes the removal of carbon-containing molecules (… such VOA), that is correlated with the drop of calorific value. Data of table 2 does not confirm this claim. For instance, comparing after hydrolysis and after 900 kPa experiment, the calorific value goes decreased from 21.9 to 10.5 kJ/mL while conversely VOA increased from 23.5 to 30.8 g/L. Could author’s comment?
Author Response
Dear reviewer, the responses to your comments are attached in a Word file. Thank you very much for your important input. Sincerely, Rok Mihelič, correspondent for the team of authors

Round 2
Reviewer 2 Report
I read and I approved the rebuttal of the authors. They had revised their manuscript. I still have some minor changes and suggestions.
L22 There is a typo in 'alkalis' please resolve.
L83-93 Did the authors use inoculum for their fermentation studies? If yes, information about the inoculum should be integrated into the manuscript.
L 97 The authors mentioned about pilot scale trial but the results of this trial cannot be followed in results and discussion. The authors need to clarify this issue.
L112-130 These two subsections should switch position for clear logical flow.
Table 2 line 1 It seems there is a mistake in standard deviations. Magnitude is higher than the average.
L224 The sentence is grammatically incorrect.
L 263-265 This statement needs a citation.
L 271- 279 This part must be moved to discussion
Response 14 of the rebuttal must be incorporated into the manuscript as a discussion.
Author Response
Dear Reviewer, we rearranged the chapters and changed the text accordingly to your requests. Please see the attached Word file with our answers, and the new paper version.
Thank you, Rok Mihelič

Reviewer 3 Report
Most of the remarks have been answered by the authors, thanks.
An error remains at line 180, please write 22.56 kJ/g SW instead of 22,560 kJ/g SW
However, I am still doubting about some data given in Table 2. Comparing DSW after hydrolysis and after 900 kPa ammonia stripping, there is a drop of energy of 11.32 kJ/mL. Considering an average of 20 kJ/g for VOA (15 for acetic acid, 20 for propionic acid and 25 for butyric acid), the energy drop corresponds to a loss of 566 gVOA /L, which is huge and higher than the fat content of DSW. Could authors solve this question?
Author Response
Dear Reviewer,
Thank you very much for your important contribution. We had to go deeply again into the measured results with repetitions. Upon your point: “I am still doubting about some data given in Table 2. Comparing DSW after hydrolysis and after 900 kPa ammonia stripping, there is a drop of energy of 11.32 kJ/mL. Considering an average of 20 kJ/g for VOA (15 for acetic acid, 20 for propionic acid and 25 for butyric acid), the energy drop corresponds to a loss of 566 g VOA /L, which is huge and higher than the fat content of DSW.”, we decided to remove our data about the gross energy value after pressure ammonia stripping. They are most probably not reliable. Unfortunately, we cannot repeat the measurements, since the samples, which were deep frozen, were destroyed when the laboratory was hand over to another owner last year. That is why we see as the only possible option to remove gross energy data from the Table 2 and from the results and discussion. Hopefully, this does not weaken the paper too much.
Sincerely, Rok Mihelič

Round 3
Reviewer 3 Report
Thank you for integrating and responding to remarks and comments